# LONGHORN: STATE SPACE MODELS ARE AMORTIZED ONLINE LEARNERS

**Bo Liu[◇,†], Rui Wang[‡], Lemeng Wu[◇], Yihao Feng[†], Peter Stone[†,⋆], Qiang Liu[†]**
[†]The University of Texas at Austin, [◇]Meta, [‡]Helixon, [⋆]Sony AI.
{lbo,lmwu}@meta.com, {yihao,pstone,lqiang}@cs.utexas.edu

## ABSTRACT

Modern large language models are built on sequence modeling via next-token prediction. While the Transformer remains the dominant architecture for sequence modeling, its quadratic decoding complexity in sequence length poses a major limitation. State-space models (SSMs) present a competitive alternative, offering linear decoding efficiency while maintaining parallelism during training. However, most existing SSMs rely on linear recurrence designs that appear somewhat ad hoc. In this work, we explore SSM design through the lens of online learning, conceptualizing SSMs as meta-modules for specific online learning problems. This approach links SSM design to formulating precise online learning objectives, with state transition rules derived from solving these objectives. Based on this insight, we introduce a novel deep SSM architecture, Longhorn, whose update resembles the closed-form solution for solving the online associative recall problem. Our experimental results show that Longhorn outperforms state-of-the-art SSMs, including the Mamba model, on standard sequence modeling benchmarks, language modeling, and vision tasks. Specifically, Longhorn achieves a **1.8x** improvement in sample efficiency compared to Mamba, and can extrapolate over contexts that are up to **16x** longer during inference. The code is provided at `https://github.com/Cranial-XIX/Longhorn`.

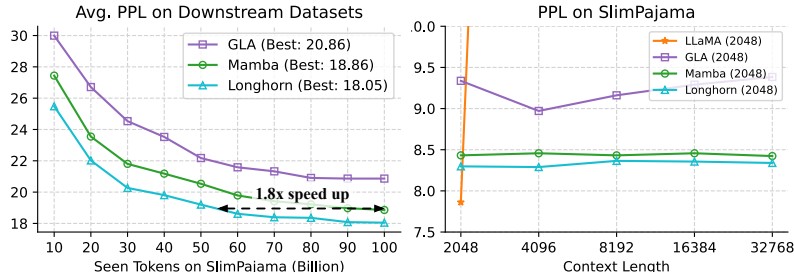

Figure 1: **(left)** The average perplexity on eight downstream datasets for GLA, Mamba, and Longhorn (1.3B model) over seen tokens on SlimPajama. Longhorn leads to a 1.8x speed up in sampling efficiency. **(right)** Longhorn, pretrained with 2048 context length, extrapolates up to 16x longer context at inference.

# 1 INTRODUCTION

The Transformer model has become the go-to architecture for sequence modeling in deep learning (Vaswani et al., 2017). However, its utility is constrained by the quadratic growth in training and decoding costs with increasing sequence length. Despite various optimizations such as efficient decoding (e.g., Chen et al., 2023; Kuperman & Dyke, 2011), KV-cache compression (e.g., DeepSeek-AI & Dai, 2024), and memory efficient implementation (e.g., Dao et al., 2022), it remains challenging to scale Transformers for autonomous and continual use with an infinite (or very long) context window.

Recent advances in linear attention models (Katharopoulos et al., 2020) and state-space models (SSMs) (Gu et al., 2021) have demonstrated their potential. These models are specialized recurrent

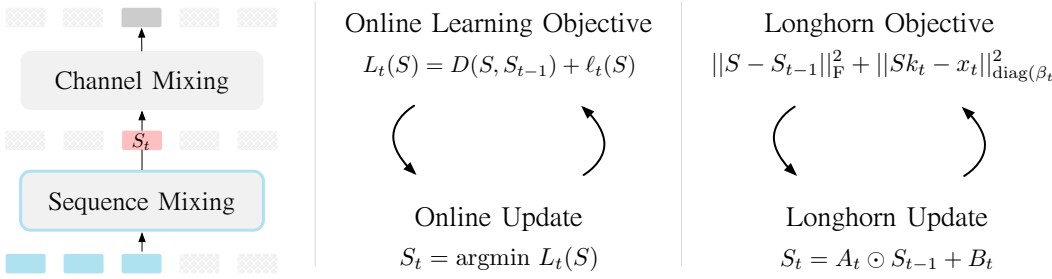

Figure 2: **(left)** Most existing sequence models consist of channel and sequence mixing layers. The sequence mixing layers can be viewed as "meta-modules" that compress history into a state $S_t$, which is then passed to later layers for sequence modeling. **(middle)** Sequence mixing can be seen as an online learning problem, where the SSM state $S_t$ optimizes an online objective. The recurrent update of $S_t$ is derived either by solving this objective in closed form or via a proximal update. **(right)** Longhorn's update solves online associative recall, where the goal is to recover $x \in \mathbb{R}^d$ based on a hint $k \in \mathbb{R}^m$ from a state matrix $S \in \mathbb{R}^{d \times m}$. Longhorn's update corresponds to the implicit online learning solution, where $A_t = 1_{d \times m} - \varepsilon_t \otimes k^{\otimes 2}$ and $B_t = (\varepsilon \odot x_t) \otimes k_t$, and $\varepsilon_t = \beta_t/(1 + \beta_t k_t^\top k_t)$. See the details in Section 3 and Algorithm 1.

neural networks capable of efficiently computing outputs in parallel when input tokens are provided simultaneously during training, thus avoiding the inefficiencies of traditional backpropagation through time. During inference, the recurrent form is employed, resulting in linear decoding efficiency. Initially, these models underperformed compared to Transformers. However, recent SSMs (e.g., Gu & Dao, 2023; Yang et al., 2023; Peng et al., 2024; De et al., 2024; Beck et al., 2024) have achieved performance parity with Transformers in language modeling tasks. Despite extensive research into various design aspects of SSMs, a guiding principle for designing SSMs remains elusive.

In this work, we propose one potential principle. We observe that one can view SSMs (or any sequence mixing layers) as "meta modules" that compress the history *online* into a memory state which is then used by later layers in the network for sequence modeling. From this perspective:

*The recurrent form of SSMs can be viewed as solving an online learning problem.*

As a result, **we can draw inspiration from online learning and confine the design choices of SSMs to reflect those learning dynamics that solve specific online prediction problems**. The aim is that, by optimizing for the right objective, the model can achieve superior performance with fewer parameters or reduced computational costs. Furthermore, this online learning perspective may offer deeper insights into the function of SSM layers in large models. In particular, the recurrent update (i.e., state-transition dynamics) of an SSM can be interpreted as a proximal update step or a closed-form solution to an online learning objective. We outline the corresponding objectives for several existing SSMs in Table 5. One significant advantage of viewing SSMs through the lens of online learning is their ability to adapt post-training during deployment, allowing them to process arbitrarily long data sequences at inference time.

Based on this insight, we propose a simple yet effective architecture (Longhorn), derived from the implicit closed-form update of an online associative recall problem. The closed-form update naturally leads to a stable recurrent form without a manually designed gating mechanism, automatically balancing *forgetting* and *learning*. Thus Longhorn does not need a separately parameterized forget gate, which saves parameters when the state size is large. We demonstrate that Longhorn performs comparably to or better than state-of-the-art SSMs like Mamba (Gu & Dao, 2023) on synthetic and large-scale sequence modeling tasks. In particular, Longhorn outperforms Mamba at the size of 1.3B-parameter when trained on 100B tokens from the SlimPajama dataset (Soboleva et al., 2023). To summarize, our contributions are:

**1) Theoretical Framework:** We propose a novel framework that views SSMs' recurrent update as solving online learning objectives. As a result, the design of SSMs reduces to the design of the online learning objectives. In particular, we introduce a novel, simple, and effective SSM, named *Longhorn*, that explicitly solves an online associative recall problem. Longhorn's recurrent update is obtained by the *closed-form* solution to the online learning objective. Consequently, Longhorn does not require a separately parameterized forget gate that appears in most existing SSMs.

---

**Algorithm 1** Longhorn's Single-layer SSM Recurrence (Inference Time)

---

1: **Parameters:** $W_q \in \mathbb{R}^{m \times d}, W_k \in \mathbb{R}^{m \times d}, W_\beta \in \mathbb{R}^{d \times d}$, where $W_\beta$ can be low-rank, horizon $T$.
2: Initialize the memory state $S_0 \leftarrow 0^{d \times m}$.
3: **for** $t \in \{1, \ldots, T\}$ **do**
4:     **1)** Receive input $x_t \in \mathbb{R}^d$.
5:     **2)** Compute the query $q_t$, key $k_t$ and $\beta_t$:

$$q_t = W_q x_t \in \mathbb{R}^m, \qquad k_t = W_k x_t \in \mathbb{R}^m, \qquad \beta_t = \text{Sigmoid}(W_\beta x_t) \in (0,1)^d.$$

6:     **3)** Update the memory state $S_t \in \mathbb{R}^{d \times m}$ via

$$S_t = \left(1_{d \times m} - \varepsilon_t \otimes k_t^{\odot 2}\right) \odot S_{t-1} + \left(\varepsilon_t \odot x_t\right) \otimes k_t, \quad \varepsilon_t = \beta_t / (1 + \beta_t k_t^\top k_t) \in (0,1)^d.$$

7:     **4)** Compute the output $o_t = S_t q_t \in \mathbb{R}^d$.
8: **end for**
9: **Note:** $\odot$ elementwise product and $\otimes$ is outer product. $x_t$ in practice is preprocessed through a linear projection followed by a Conv1d operation as in Mamba (Gu & Dao, 2023).

---

**2) Empirical Results:** Longhorn demonstrates better performance than existing SSMs including Mamba, across both synthetic associative recall tasks and the large-scale language modeling task. Moreover, it achieves 1.8x improvement in sample efficiency compared to Mamba (See Figure 1 (left)). Longhorn's training speed is as fast as Mamba, as we only replace the SSM module in the Mamba architecture with Longhorn's recurrence. So it serves as a drop-in replacement for Mamba. Lastly, Longhorn, trained with 2048 context length can extrapolate to 32K context length at inference time without much perplexity drop (See Figure 1 (right)).

**Notation** Throughout this work, we use $\odot$ to denote the Hadamard (elementwise) product, and $\otimes$ to denote the Kronecker (or outer) product between two tensors. Uppercase letters $A$, $B$, etc. denote matrices, while lowercase $k, v$ are in general vectors. $\|\cdot\|$ by default refers to the $\ell_2$ norm for vectors.

## 2   BACKGROUND

In this section, we provide a brief introduction to contemporary deep state space models (deep SSMs).

Modern large language models are sequence-to-sequence models consisting of a stack of layers $\boldsymbol{y} = \Phi_L \circ \cdots \circ \Phi_1(\boldsymbol{x})$ that sequentially processes an input sequence $\boldsymbol{x} = \{x_t\}_{t=1}^T$, where $T$ is the context length. Specifically, transformers consist of alternative stacks of self-attention (SA) and multi-layer perceptron (MLP) layers that conduct *mixing* (i.e., information aggregation) on the sequence and channel dimensions, respectively.

Deep SSMs replace the SA layers with SSM layers. Some variants of SSM models leave the MLP layers unchanged (Sun et al., 2023; Yang et al., 2023; De et al., 2024), while others fuse the SSM layer and the MLP layer into a single unified module (Gu & Dao, 2023). But in both cases, the sequence mixing is done by the SSM module, and the channel mixing is done by the channel-wise MLP. Taking Mamba as an example (Gu & Dao, 2023), a Mamba model consists of a stack of homogeneous modules named Mamba block (the $\Phi_i(\boldsymbol{x})$); we provide a visualization of a single Mamba block in Figure 3 (Gu & Dao, 2023), which consists of an SSM block for sequence mixing (red), and an MLP block for channel mixing (blue).

**SSM: General Form** The SSM block (in red) plays the crucial role of sequence mixing. It works by iteratively updating a memory state matrix $S_t \in \mathbb{R}^{d \times m}$ with a linear recurrence:

$$S_t = A(x_t) * S_{t-1} + B(x_t), \qquad \forall t \in \{1, \ldots, T\}, \qquad S_0 = 0, \qquad (1)$$

where $x_t$ is the input at time $t$, $S_t$ is the model's *state*, $A_t, B_t \colon \mathbb{R}^d \to \mathbb{R}^{d \times m}$ are some functions of the input and $*$ is a multiplication operation of choice, such as Hadamard product or matrix product.

Given the state $S_t$, SSMs often give the output token at the next layer via a gated linear unit (GLU) (Dauphin et al., 2017):

$$y_t = \mathtt{Readout}(S_t, x_t) = W_1\big(o_t \odot \sigma(W_2 x_t)\big), \qquad o_t = C(x_t)S_t,$$

where we first get $o_t$ via a state-dependent linear projection on $S_t$, which is then fed into a subsequent channel mixing gated linear unit (blue in Figure 3), where $\sigma(\cdot)$ is a non-linear activation function.

A key feature of this design in Equation 1 is that $S_t$ has a linear recurrence, i.e., $S_t$ is linear in $S_{t-1}$. Crucially, this allows us to express all $S_t$ in an explicit form that can be calculated in parallel: when all $\boldsymbol{x} = \{x_t\}_t$ are available as in the training phase, $\{S_t\}_t$ can be written into

$$S_t = \sum_{t' \leqslant t} (\overline{A}_{t' \to t}) B(x_{t'}), \quad \text{where} \quad \overline{A}_{t' \to t} = \prod_{t' < \tau \leqslant t} A(x_\tau). \quad (2)$$

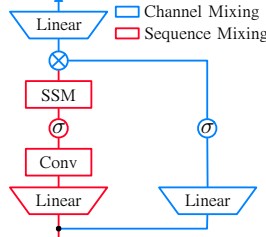

Figure 3: Mamba Block

Here $\prod$ denotes the product induced by multiplication operator $*$. The resulting cumulative product $\overline{A}_{t' \to t}$ can be implemented efficiently in parallel with the prefix scan algorithm (e.g., Harris et al., 2007), which only requires $\mathcal{O}(\log T)$ ($T$ is the sequence length) parallel operations. From now on, we will abbreviate $A(x_t)$ and $B(x_t)$ as $A_t$ and $B_t$, respectively.

**Designs of** $(A_t, B_t, *)$   Existing variants of SSMs mainly differ in the design choices of the networks $A_t, B_t$, and the associated operator $*$ in the linear recurrence. A core issue here is that the memory state $S_t \in \mathbb{R}^{d \times m}$, designed to be $m$ times the input $x_t$ in size, must be as large as possible to maintain sufficient information during recurrence. This makes the architecture design of $A_t, B_t$, both mapping $\mathbb{R}^d$ to $\mathbb{R}^{d \times m}$ challenging. A naive linear mapping would result in $d \times d \times m$ weights, which is prohibitively large. This makes it necessary to impose certain low-dimensional structures in $A_t, B_t$, which is the main difference from existing designs of SSMs. In Appendix A, we summarize some existing deep SSM models in the form of Equation 1.

## 3    AN ONLINE LEARNING PERSPECTIVE FOR SEQUENCE MIXING

As demonstrated in the previous section, designing a state-space model (SSM) depends on the specific selection of $(A_t, B_t, *)$, which is intricate and somewhat artisanal. In this section, we propose to streamline SSM design through an online learning perspective. The main idea is to treat the SSM layers as learning modules that learn to compress information along the sequence dimension. From this perspective, the SSM layers are *learning to learn*, such that during the inference time, these layers are still learning (compressing) new information online.

We begin with an overview of online learning and subsequently demonstrate how SSM can be framed as an online learning problem. Finally, we present a straightforward architecture based on the closed-form solution of the implicit online learning algorithm.

### 3.1    SSM AS ONLINE LEARNING

We advocate viewing the recurrence of SSM as solving an online learning problem. In online learning, the agent picks a state $s_t$ at time $t$ and then incurs a loss $\ell_t(s_t)$. The goal is to minimize

$$\min_{\{s_t\}} \sum_t \ell_t(s_t). \quad (3)$$

For instance, consider online linear prediction, where at each step the agent is given an input-label pair $(x_t, y_t)$ and $\ell_t(s_t) = \frac{1}{2}||s_t^\top x_t - y_t||^2$ is the $\ell_2$ regression loss, then the problem becomes an online regression problem, and the goal is to successfully predict $y_t$ given $x_t$ at future time steps, with the key feature that the prediction $(s_t)$ can change with each new data point.

Online convex programming (OCP) (e.g., Zinkevich, 2003) yields a principled approach to solving Equation 3 when $\ell_t$ are convex, by trading-off the "stability" and "plasticity" (e.g., Mermillod et al.,

2013). Formally, an online convex programming algorithm updates $s_t$ by solving a regularized cost function:

$$s_t = \arg\min_s L_t(s), \qquad L_t(s) = \underbrace{D_\phi(s, s_{t-1})}_{\text{stability}} + \underbrace{\beta_t \ell_t(s)}_{\text{plasticity}}, \tag{4}$$

where $\beta_t \in \mathbb{R}^+$ and $D_\phi$ is a discrepancy measure, often a Bregman divergence induced by the convex function $\phi$ (e.g., when $\phi(x) = \frac{1}{2}\|x\|^2$, $D_\phi(s, s_{t-1}) = \frac{1}{2}\|s - s_{t-1}\|^2$). Here the first term ensures the updated $s$ will be close to the previous $s_{t-1}$, so the agent suffers less from *catastrophic forgetting*, while the second term ensures the agent is incorporating new knowledge from minimizing the new loss $\ell_t(s)$. Hence, $\beta_t$ controls the trade-off between stability and plasticity.

## 3.2 THE LONGHORN ARCHITECTURE

Under the online learning framework, the **design of an SSM reduces to the design of $D_\phi$ and $\ell_t$** in Equation 4. This provides a unified framework for the existing SSM variants. We summarize in Table 5 in Appendix B the online learning interpretation of several existing SSM architectures.

In this work, we explore a highly simplified and natural design called *Longhorn* guided by the online principle (see the last row of Table 5). In particular, we consider $\{(k_t, x_t)\}_t$ as the input stream, where $k_t \in \mathbb{R}^m$ and $x_t \in \mathbb{R}^d$ are the key-value pairs, just as in the Transformer model (Vaswani et al., 2017). In practice, as in Mamba (Gu & Dao, 2023), $k_t = W_k x_t \in \mathbb{R}^m$, where $W_k \in \mathbb{R}^{m \times d}$, is a linear mapping from $x_t$.

We want to recurrently update hidden states $\{S_t\}_t$, where $S_t \in \mathbb{R}^{d \times m}$ is a matrix that summarizes the information up to time $t$. We posit the following OCP objective for updating $S_t$:

$$S_t = \arg\min_{S \in \mathbb{R}^{d \times m}} \left\{ \|S - S_{t-1}\|_{\text{F}}^2 + \|Sk_t - x_t\|_{\text{diag}(\beta_t)}^2 \right\}. \tag{5}$$

Here, $\|\cdot\|_{\text{F}}$ denotes the Frobenius norm of a matrix, $\beta_t \in \mathbb{R}^d$ is a vector controlling how much new information about $x_t$ we want the model to incorporate for $S_t$. For instance, $\beta_{t,i} = 0$ implies $S_{t,i} = S_{t-1,i}$ (i.e., the $i$-th row of $S$ remains unchanged), while a large $\beta_{t,i}$ implies the model empties some part of $S_i$ for incorporating $x_{t,i}$.

From a high-level perspective, Equation 5 is solving an online prediction problem of learning a weight matrix $S$ to predict $x_t$ given $k_t$ with a linear model $x_t \approx S^\top k_t$. It is a supervised formulation of the *associative memory* problem of memorize $(k_t, x_t)$ pairs by learning a mapping from $k_t$ to $x_t$, such that given a key (input) $k_t$ the model can retrieve (predict) its corresponding value (label) $x_t$.

The objective in Equation 5 is motivated by the observation that the self-attention layer of the Transformer exhibits a form of online associative recall (often referred to as the induction head property) (Olsson et al., 2022). This capability has been shown to underpin the model's ability to perform in-context learning (Brown, 2020). To explain the connection, in-context learning refers to the model's ability, during inference, to generalize from a set of provided $(k, x)$ (question-answer) pairs and apply this understanding to a new question. This closely parallels associative recall, where the model retrieves relevant information from past interactions to address new inputs.

Fortunately, this simple objective gives a closed-form solution for $S_t$, which coincides with the implicit online learning method (e.g., Kulis & Bartlett, 2010), according to Theorem 3.1 (We provide the proof in Appendix C):

**Theorem 3.1.** *The closed form solution for $S_t$ for objective in Equation 5 is*

$$S_{t,i} = (I - \varepsilon_{t,i} k_t k_t^\top) S_{t-1,i} + \varepsilon_{t,i} k_t x_{t,i}, \quad \text{where } \varepsilon_{t,i} = \frac{\beta_{t,i}}{1 + \beta_{t,i} k_t^\top k_t} \in [0, \infty). \tag{6}$$

Here, $S_{t,i}$ refers to the $i$-th row of $S_t$, $\beta_{t,i}$ refers to the $i$-th element of $\beta_t$. As $k_t k_t^\top$ is a matrix, it is hard to compute its cumulative product for conducting a parallel scan. As a result, in practice, we use the diagonal approximation $1_m - \varepsilon_{t,i} k_t^{\odot 2}$ in place of $I - \varepsilon_{t,i} k_t k_t^\top$, where $a^{\odot 2} = a \odot a$ and $1_m$ is the $m$-dimensional all-one vector. Following Mamba (Gu & Dao, 2023) and Transformer (Vaswani et al., 2017), we make $k_t = W_k x_t \in \mathbb{R}^m$ and $\beta_t = \sigma(W_\beta x_t) \in \mathbb{R}^d$ (both are functions of $x_t$), where

the activation $\sigma$ (the Sigmoid function) is to ensure that $\beta_t$ is positive and bounded. In summary, the final Longhorn update of $S_t$ becomes:

$$S_t = A_t \odot S_{t-1} + B_t, \quad \text{where} \quad A_t = (1_{d \times m} - \varepsilon_t \otimes k_t^{\odot 2}), \quad B_t = (\varepsilon_t \odot x_t) \otimes k_t. \quad (7)$$

The final architecture of Longhorn follows Mamba strictly (Figure 3), except that we replace the SSM block with Longhorn's recurrence. We also provide an efficient CUDA kernel for it. The full inference-time algorithm is provided in Algorithm 1. One can compare Equation 7 to Equation 8 and other SSMs in Appendix A. Longhorn does not introduce an extra "forgetting" gate (hence it has fewer parameters), because the forgetting gate is naturally derived from the key vector, i.e., $(1_{d \times m} - \varepsilon_t \otimes k_t^{\odot 2})$.

**Advantages of Longhorn**

1. While we can derive the learning objective for some of the existing SSMs, Longhorn is the first SSM designed for explicitly solving an online regression problem.

2. Longhorn does not require a specific forget gate (e.g., $\alpha_t$ in GLA or $A$ matrix in Mamba). The forgetting is naturally linked to the key vector $k_t$ through the derivation. This saves about $\mathcal{O}(d \times m)$ parameters per SSM module, where $m$ is the dimension of $k_t$, and $d$ is the dimension of $x_t$. However, Longhorn demonstrates better performance even with fewer parameters than Mamba (See Figure 1 (left), Table 2, Table 3).

3. The closed-form solution in Equation 6 **does not need any specific initialization**. In contrast, Mamba requires a special careful initialization of the $A$ and $\varepsilon_t$.

4. Unlike DeltaNet (Yang et al., 2024), which struggles to extrapolate beyond training contexts, Longhorn successfully extrapolates to contexts **16x longer** than it was trained for (Figure 1 (right)).

## 4 RELATED WORK

This section provides a summary of recent advances in linear attention and state space models.

**Linear Attention Models**    Several methods reduce the quadratic complexity of Transformers by making attention linear with respect to context length. Linformer projects keys and values into a constant-size matrix, bypassing the scaling with sequence length (Wang et al., 2020). Linear Transformer replaces the Softmax function with a decomposable similarity function, achieving linear complexity (Katharopoulos et al., 2020). Performer approximates softmax attention using orthogonal random features (Choromanski et al., 2020). RetNet adds constant forgetting and rotation (Sun et al., 2023), while Gated Linear Attention introduces learnable forget gates (Yang et al., 2023). Linear attention can also be seen as a fast weight network where a slow net adapts a fast network's parameters online using inputs (Schlag et al., 2021).

**State Space Models**    State space models (SSMs) focus on parallelizable linear recurrent networks. Initially, a constant state transition matrix $A$ allows recurrence to be computed via convolution (Li et al., 2022; Gu et al., 2021). Key models include Diagonal State Space (DSS) (Gupta et al., 2022), Gated State Space (GSS) (Mehta et al., 2022), S5 (Smith et al., 2022), Bidirectional Gated SSM (BiGS) (Wang et al., 2022), H3 (Fu et al., 2022), and Mamba (Gu & Dao, 2023). Efficient recurrent networks often resemble SSMs, such as Deep Linear Recurrent Units (LRUs) (Orvieto et al., 2023; De et al., 2024), Hierarchically Gated Linear RNNs (HGRN) (Qin et al., 2024b;a), and RWKV (Peng et al., 2023; 2024).

**Fast Weight Programmer**    The idea of networks modifying their own weights in response to inputs dates back to the Fast-weight Programmer (Schmidhuber, 1992; 1993; Schlag & Schmidhuber, 2017; Schlag et al., 2021). These models update a weight matrix $W \in \mathbb{R}^{d \times m}$ via the outer product of two vectors: $\Delta W = x_t \otimes k(x_t)$, a mechanism similar to Linear Attention. Our framework extends this concept by adapting the weight update process to suit specific online learning objectives, enhancing its use in dynamic learning environments. More recently, Transformers (self-attention) has also been discovered as conducting mesa-optimization, adjusting the model as it reads new inputs (Von Oswald et al., 2023).

**Concurrent Work** Two concurrent works share similar ideas with ours. Yang et al. (2024) propose a chunk-wise parallel approach to scale DeltaNet (Schlag et al., 2021) for large-scale language modeling. DeltaNet's update rule, which is viewed as a gradient step for an online regression objective, results in a state transition matrix $A(x_t) = (I - \beta_t k_t k_t^\top)$, which can have eigenvalues $>1$, leading to instability. To address this, Yang et al. (2024) normalize the key vector $k_t$ by its $\ell_2$ norm, which can be restrictive. In contrast, Longhorn ensures stability with a closed-form update, using a diagonal approximation ($k_t^{\odot 2}$), allowing for both parallel scan (as in Mamba) and chunk-wise parallel training (as in GLA), making it as fast as existing SSMs. Additionally, we provide a parallel scan CUDA kernel, enabling Longhorn to serve as a drop-in replacement for Mamba. Sun et al. (2024) introduce the Test-Time Training framework, where state updates are derived from a gradient step on an online regression objective. To maintain parallelism, they assume that each gradient step at $x_t$ uses the initial state $s_0$, enabling matrix multiplication. In contrast, Longhorn computes the closed-form solution for the *every* token, offering greater flexibility.

## 5 EXPERIMENTS

We validate Longhorn's performance through the following experiments:

**1)** We compare Longhorn against other SSMs on the multi-query associative recall benchmark (Arora et al., 2023a) and find that **Longhorn is the only model to achieve near-perfect recall at sequence lengths up to 512 with a hidden dimension of 64**. We further compare Longhorn (1B) against Mamba (1B) on real-world recall-intensive tasks (Arora et al., 2024a;c) and find that Longhorn also achieves a better recall rate compared against Mamba.

**2)** Using the OpenWebText dataset (Gokaslan & Cohen, 2019), we assess Longhorn's performance on language modeling with model sizes of 120M and 350M, and context lengths of 1024 or 4096, showing **it consistently outperforms other SSMs in validation perplexity**.

**3)** We train a 1.3B language model on the SlimPajama dataset (Soboleva et al., 2023) with 100B tokens and compare its performance across 8 benchmarks, where **Longhorn achieves better final performance and $>$1.8x better sample efficiency than Mamba and GLA**.

**4)** We additional apply Longhorn to vision domain and compare it against the Vision Mamba (ViM) (Zhu et al., 2024) model (Appendix 5.6), where **Longhorn achieves performance comparable (slightly superior) to that of the ViM model**.

### 5.1 MULTI-QUERY ASSOCIATIVE RECALL

We first consider the synthetic benchmark Multi-Query Associative Recall (MQAR) (Arora et al., 2023a). The agent observes a sequence of tokens $\{k_1, v_1, k_2, v_2, \ldots, k_T, v_T\}$, where each consecutive two-tokens become a key-value pair. At test time, the agent is provided with multiple $k \sim \{k_1, \ldots k_T\}$, the goal is to "retrieve" the corresponding values. Following the original benchmark, we consider the sequence length $T \in \{64, 128, 256, 512\}$ and model dimension (size of the latent embedding of a token) $d \in \{64, 128, 256, 512\}$. We compare against 1) Transformer model (Attention), 2) Based architecture, which combines an SSM with local-attention, where the SSM is derived from the Taylor approximation of the self-attention (Arora et al., 2024b), 3) Hyena (Poli et al., 2023), which is a special SSM that adopts long convolution via fast fourier transform, 4) RWKV (Peng et al., 2023), which can be viewed as the division of two SSMs (i.e., $y = a/b$, where $a, b$ are outputs from two SSMs). The state-transition matrix is a scalar, 5) BaseConv (Arora et al., 2023a), an SSM that combines linear projection with convolution, and 6) Mamba (Gu & Dao, 2023), the state-of-the-art SSM that has data-dependent $A$ and $B$ (See equation 8). Each experiment individually searches for the best learning rate from $\{10^{-4}, 4.6 \times 10^{-4}, 2.2 \times 10^{-3}, 10^{-2}\}$. Results are summarized in Figure 4.

**Observation:** From the figure, we can see that Longhorn, which is designed to perform the associative recall task by solving the online prediction objective, outperforms existing SSM variants even at the sequence length of 512 and a small model dimension of 64.

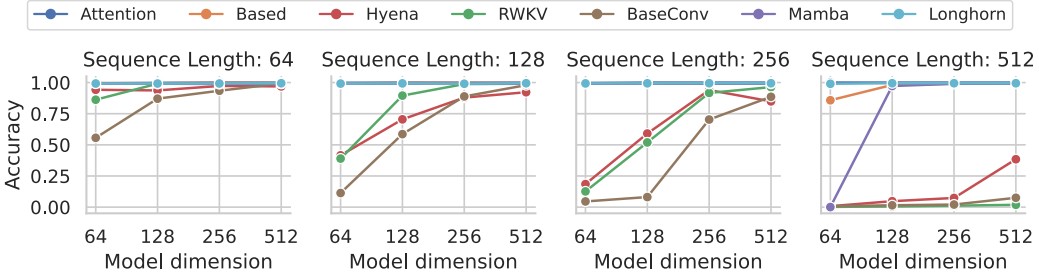

Figure 4: Comparison of Longhorn to state-of-the-art SSMs on the MQAR benchmark. y-axis is the recall rate.

## 5.2 REAL-WORLD RECALL INTENSITIVE TASK

To further evaluate Longhorn's ability to recall in long real-world sequences, following the experiment setup in Arora et al. (2024c), we consider the following six recall-intensive tasks: information extraction tasks like FDA and SWDE (Arora et al., 2024a; 2023b; Wu et al., 2021; Deng et al., 2022), and question-answering benchmarks like Squad (Rajpurkar et al., 2018), NaturalQuestion (NQ) (Kwiatkowski et al., 2019), TriviaQA (Joshi et al., 2017), and DROP (Dua et al., 2019). Following Arora et al. (2024c), we report both the vanilla question answering accuracy, and the accurayc under the JRT-Prompt format (Arora et al., 2024c), where the context is repeated twice before the model conducts the final completion. The zero-shot prompt includes up to 1k tokens in the input and JRT-Prompt includes up to 2k tokens in the input for all tasks, as it repeats the context twice. The 1B checkpoint of Longhorn is taken from Section 5.4. The results are provided in the following table:

| Model | FDA | SWDE | Squad | NQ | TriviaQA | DROP | Average |
|---|---|---|---|---|---|---|---|
| Mamba-1.3B | 33.2 / 40.6 | **35.0** / 36.2 | 26.6 / 32.6 | **37.4** / 52.7 | 56.3 / **56.9** | 20.4 / 31.5 | **34.8** / 41.8 |
| Longhorn-1.3B | **40.2 / 50.4** | 33.2 / **42.3** | **27.6 / 33.2** | 35.0 / **55.0** | **58.5** / 55.9 | **21.3 / 33.3** | 36.0 / **45.0** |

Table 1: Longhorn and Mamba's recall performance across six real-world recall-intensive benchmarks.

**Observation:** From the table, it shows that Longhorn 1.3B achieves a 3.4%/7.7% improvement of recall accuracy under the vanilla/JRT-Prompt context format, compared against the same size Mamba.

## 5.3 SCALING LAW ON OPENWEBTEXT

In this section, we consider language modeling tasks on models with 120M or 350M parameters with 1024 or 4096 context length. We choose the OpenWebText dataset as it is small and serves as an easily accessible benchmark for quick benchmarks.[1] The details about the architecture is provided in Appendix D. We consider the following baseline models: LLaMA (Touvron et al., 2023), RetNet (Sun et al., 2023), Mamba (Gu & Dao, 2023), RWKV (Peng et al., 2023), and GLA (Yang et al., 2023). Then we experiment with 1024 or 4096 context length $T$ and model sizes around 120M or 350M. Results are summarized in Table 2 and Figure 5.

**Observation:** From the figure and table, we can see that Longhorn consistently outperforms baseline SSMs up to 350M and 4096 context length.

## 5.4 LARGE-SCALE LANGUAGE MODELING

For the large-scale language modeling task, we followed the GLA (Yang et al., 2023) setup, training a 1.3B parameter model on the SlimPajama (Soboleva et al., 2023) dataset with 100B tokens and a batch size of 2M. We used the AdamW optimizer (Loshchilov & Hutter, 2017) with a weight decay of 0.01, cosine learning rate decay (peak: $3e - 4$, final: $3e - 5$), and gradient clipping of

---

[1]We adapted code from the nanoGPT repository https://github.com/karpathy/nanoGPT, which is a minimal reproduction of GPT-2 model using PyTorch.

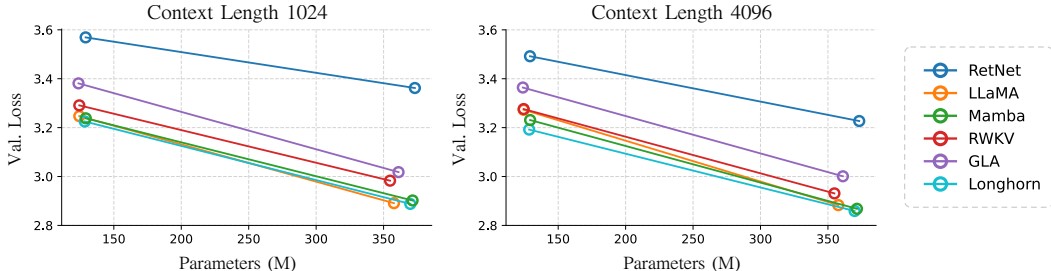

Figure 5: Scaling law with 1024 and 4096 context length on OpenWebText with various SSM models and the LLaMA (strong Transformer) baseline.

| Model | # Param. (M) | Val. Loss (↓) | | # Param. (M) | Val. Loss (↓) | |
|---|---|---|---|---|---|---|
| | | $T = 1024$ | $T = 4096$ | | $T = 1024$ | $T = 4096$ |
| RetNet | 129.1 | 3.569 | 3.492 | 373.2 | 3.362 | 3.227 |
| GLA | 123.8 | 3.381 | 3.364 | 361.1 | 3.018 | 3.001 |
| RWKV | 124.4 | 3.291 | 3.276 | 354.8 | 2.983 | 2.931 |
| Mamba | 129.2 | 3.238 | 3.231 | 371.5 | 2.902 | 2.868 |
| LLaMA | 124.4 | 3.247 | 3.273 | 357.7 | 2.891 | 2.883 |
| Longhorn | 128.6 | **3.225** | **3.192** | 369.8 | **2.888** | **2.859** |

Table 2: Language modeling scaling law against LLaMA (Touvron et al., 2023), RetNet (Sun et al., 2023), RWKV (Peng et al., 2023), and Mamba (Gu & Dao, 2023). All models are trained on the OpenWebText dataset (Gokaslan & Cohen, 2019). Models vary from 120-350M parameters and 1024-4096 context length.

1.0. Comparisons were made against LLaMA, Mamba, and GLA models (context size: 2048). We evaluated on eight standard downstream tasks, including PIQA (Bisk et al., 2020), HellaSwag (Hella) (Zellers et al., 2019), WinoGrande (Wino) (Sakaguchi et al., 2021), ARC-easy (ARC-e) and ARC-challenge (ARC-c) (Clark et al., 2018), OpenBookQA (OBQA) (Mihaylov et al., 2018), Social Interaction QA (SIQA) (Sap et al., 2019), and Boolean questions (BoolQ) (Clark et al., 2019). We report the average perplexity across the above eight datasets throughout training in Figure 1 (left). Then we summarize the downstream evaluation results in Table 3.

| Model | State Size | PIQA | Hella | Wino. | ARC-e | ARC-c | OBQA | SIQA | BoolQ | Avg. |
|---|---|---|---|---|---|---|---|---|---|---|
| | | acc ↑ | acc_norm ↑ | acc ↑ | acc ↑ | acc_norm ↑ | acc ↑ | acc_norm ↑ | acc ↑ | |
| LLaMA | 8M | 55.08 | 55.36 | 71.73 | 59.26 | 32.19 | 43.35 | 45.16 | 62.13 | 53.03 |
| GLA | 512K | 55.55 | 49.10 | 71.12 | 58.86 | 28.11 | 41.67 | 44.91 | 59.21 | 51.07 |
| Mamba | 64K | 54.21 | 53.61 | 71.67 | 61.05 | 30.15 | 43.94 | 44.18 | 59.22 | 52.25 |
| Longhorn | 64K | 55.78 | 52.30 | 71.00 | 60.63 | 29.53 | 43.55 | 44.68 | 61.29 | **52.35** |

Table 3: Language modeling results against LLaMA (Touvron et al., 2023), RetNet (Sun et al., 2023), and Mamba (Gu & Dao, 2023). All models are trained on the same subset of the SlimPajama dataset with the Mistral tokenizer. The 340M/1.3B models are trained for 15B/100B tokens respectively. **State Size** is the effective state size of an SSM per layer. For instance, GLA's state size (1024K) is computed by $md/h$, where the key and value dimensions are $m = 1024$ and $d = 2048$, and there are 4 heads $h = 4$. The individual task performance is via zero-shot. The last column shows the average value over the results on all benchmarks.

**Observation:** From Figure 1 (left), it is evident that Longhorn not only achieves a lower average perplexity but also improves sampling efficiency by **1.8x** compared to Mamba. In other words, Longhorn reaches the same average perplexity with nearly half the training data required by Mamba. From the Table 3, we can see that up to a 1.3B model, Longhorn remains strong among all baseline models and achieves slightly better result than Mamba, even though it has a bit fewer parameters.

## 5.5 ABLATION ON LENGTH EXTRAPOLATION

We evaluate how Longhorn extrapolates to a context length longer than 2048 (training context length) at inference time. In particular, we pick a disjoint validation set from SlimPajama dataset, rearrange it into batches of sequences of length $T \in \{2048, 4096, 8192, 16384, 32768\}$, and then evaluate the pretrained model's perplexity on those sequences. The results are summarized in Figure 1 (right).

**Observation:** From the figure, we observe that Longhorn successfully extrapolates to contexts up to **16x** longer than those used during training, this contrasts with DeltaNet (Yang et al., 2024), which highlights a limitation in that the model cannot extrapolate to longer contexts. In contrast, LLaMA, as a Transformer-based model, fails to extrapolate beyond its training context length.

## 5.6 VISION STATE SPACE MODELS

In addition to language tasks, recent works have also applied state space models to the vision domain, leveraging their superior training efficiency. In particular, following the Vision Mamba (ViM) (Zhu et al., 2024), we conduct experiments on the ImageNet (Deng et al., 2009) classification task. Similar to ViM, We apply a bi-directional scan with Longhorn SSM (ViL) and compare the results with ViM on both the TINY and SMALL configurations described in the ViM paper.

| Model | # Param | Top-1 Accuracy |
|---|---|---|
| ViM-Tiny | 7M | 76.1 |
| ViL-Tiny (ours) | 7M | **76.4** |
| ViM-Small | 26M | 80.5 |
| ViL-Small (ours) | 26M | **80.7** |

Table 4: Top-1 Accuracy on ImageNet classification for Vision Mamba (ViM) and Vision Longhorn (ViL).

**Observation:** The results from Table 4 demonstrate that the Vision Longhorn model (ViL) achieves comparable (slightly better) performance to the original ViM. Note that we use the best hyperparameters for ViM without additional tuning, and ViL does not require two additional parameters for the forward and backward $A$ matrices, as they are computed directly based on the key $k$ vector.

## 6 CONCLUSION AND FUTURE WORK

This work introduces a novel approach to designing deep state-space models (SSMs) by conceptualizing the recurrence update as solving an online objective. Based on this, we propose Longhorn, a novel SSM model that explicitly solves online associative recall in closed form. Longhorn is parallelizable and achieves state-of-the-art performance among SSMs on MQAR, language modeling, and image classification tasks. One future direction is to explore other online learning objectives. Additionally, recent studies (Ren et al., 2024) suggest that incorporating sliding-window attention with Mamba improves performance. We anticipate similar benefits for Longhorn.

## 7 ACKNOWLEDGEMENT

This work has taken place in the Learning Agents Research Group (LARG) and the Statistics & AI group at UT Austin. LARG research is supported in part by NSF (FAIN-2019844, NRT-2125858), ONR (N00014-18-2243), ARO (W911NF-23-2-0004, W911NF-17-2-0181), DARPA (Cooperative Agreement HR00112520004 on Ad Hoc Teamwork) Lockheed Martin, and UT Austin's Good Systems grand challenge. Peter Stone serves as the Executive Director of Sony AI America and receives financial compensation for this work. The terms of this arrangement have been reviewed and approved by the University of Texas at Austin in accordance with its policy on objectivity in research. The Statistics & AI group receives support in part from NSF CAREER1846421, SenSE2037267, Office of Navy Research, and NSF AI Institute for Foundations of Machine Learning (IFML).

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

## A    PRIOR DEEP STATE SPACE MODELS

**Example A.1** (Linear Attention Variants). *Linear Attention (LA) ([Katharopoulos et al., 2020](#)), Retention Network (RetNet) ([Sun et al., 2023](#)), and Gated Linear Attention (GLA) ([Yang et al., 2023](#)) all assume $A_t, B_t$ yield rank-1 (or even constant) outputs:*

$$S_t = A_t \odot S_{t-1} + v(x_t) \otimes k(x_t), \quad \text{with} \quad \begin{cases} A_t = 1 & \text{(LA)} \\ A_t = c \in [0,1] & \text{(RetNet)} \\ A_t = 1 \otimes \alpha(x_t) & \text{(GLA)} \end{cases},$$

*where $S_t \in \mathbb{R}^{d \times m}$, $v(x_t) \in \mathbb{R}^d$, $k(x_t) \in \mathbb{R}^m$ are linear mappings of $x_t$, and $\otimes$ denote the outer product. In practice, one can use $h$ heads as in the multi-head attention to save some computation, where the $m$ and $d$ dimensions are divided into $h$ groups and each group performs its own LA variant. The outer product complexity reduces to $\mathcal{O}(h * m/h * d/h = md/h)$. But then the effective size of $S_t$ also shrinks to $md/h$.*

**Example A.2** (Mamba ([Gu & Dao, 2023](#))). *The Mamba architecture is derived by discretizing a continuous linear dynamics. Its discretized update is:*

$$\begin{aligned} S_t &= A_t \odot S_{t-1} + B_t, \quad \text{where} \\ A_t &= \exp(A \odot (\varepsilon(x_t) \otimes 1)), \qquad B_t = (\varepsilon(x_t) \odot x_t) \otimes k(x_t). \end{aligned} \tag{8}$$

*where $S_t \in \mathbb{R}^{d \times m}$ with $m = 16$ by default, $\varepsilon(x_t) \in \mathbb{R}^d$, $k(x_t) \in \mathbb{R}^m$ linear mappings of $x_t$, and $A \in \mathbb{R}^{d \times m}$ is a data independent (not depending on $x_t$ trainable weight matrix.*

*In Mamba, both $A_t$ and $B_t$ depend on $\varepsilon(x_t)$, which represents the step size for the SSM update.*

*In practice, Mamba does not use multiple heads as in linear attention variants. Perhaps the main reason is that given a fixed $m$ and $d$, the largest memory state will be with $h = 1$ (as the effective size of $S_t$ is $md/h$). In addition, Mamba's output is $o_t = C(x_t)S_t + D_t \odot x_t$, which has an additional residual part $D_t \odot x_t$.*

**Example A.3** (Griffin ([De et al., 2024](#))). *In Mamba and the linear attention variants, the outer product serves as a critical role in lifting vectors to matrices. The recent Griffin architecture abandons the outer product and performs pure elementwise product:*

$$s_t = a(x_t) \odot s_{t-1} + \sqrt{1 - a(x_t)} \odot i(x_t) \odot x_t,$$

*where $s_t, a(x_t), i(x_t)$ are all $\mathbb{R}^d$. This yields smaller memory states, but in practice, Griffin is combined with local attention (i.e., the sliding-window self-attention) to strengthen its capability.*

**Example A.4** (RWKV ([Peng et al., 2023](#))). *The original RWKV also performs elementwise recurrence. It maintains a state of ratio form $s_t = u_t/z_t$, where $u_t, z_t$ are updated separately by two SSMs:*

$$\begin{aligned} s_t &= u_t/z_t \\ u_t &= \exp(-w) \cdot u_{t-1} + \exp(k(x_t)) \odot v(x_t), \qquad z_t = \exp(-w) \cdot z_{t-1} + \exp(k(x_t)), \end{aligned}$$

*where all the vectors are of size $\mathbb{R}^d$, and $w > 0$ is a* trainable weight *for controlling the forgetting. In the most recent RWKV version ([Peng et al., 2024](#)), the denominator $z_t$ is removed, and the elementwise product is replaced with the outer product, which makes it more similar to an LA variant.*

**Example A.5** (HGRN2 ([Qin et al., 2024a](#))). *The Gated Linear RNNs with State Expansion (HGRN2) model is represented with the following recurrence:*

$$S_t = (1 \otimes f(x_t)) \odot S_{t-1} + i(x_t) \otimes (1 - f(x_t)).$$

*Here, $f(x_t) \in [0,1]$ is the forget gate, $(1 - f(x_t))$ is the input gate, and $i(x_t)$ is the input vector. HGRN2 thus resembles an RNN.*

## B    EXISTING STATE SPACE MODELS' ONLINE OBJECTIVES

We reverse-engineer some existing deep SSMs' online learning objectives in Table 5.

| Method | Online Learning Objective $L_t(s)$ (assume $x_t \in \mathbb{R}$) | Online Update |
|---|---|---|
| LA | $\|S - S_{t-1}\|_F^2 - 2\langle Sk_t, x_t\rangle$ | $S_t = S_{t-1} + x_t \otimes k_t$ |
| RetNet | $\gamma\|S - S_{t-1}\|^2 + (1-\gamma)\|S\|_F^2 - 2\langle Sk_t, x_t\rangle$ | $S_t = \gamma S_{t-1} + x_t \otimes k_t$ |
| GLA | $\|S - S_{t-1}\mathrm{diag}(\alpha_t)\|_F^2 + 2\langle Sk_t, x_t\rangle$ | $S_t = S_{t-1}\mathrm{diag}(\alpha_t) + x_t \otimes k_t$ |
| Griffin | $\left\|\sqrt{\alpha_t} \odot (s - s_{t-1})\right\|^2 + \left\|\sqrt{1-\alpha_t} \odot s\right\|^2 - 2\sqrt{1-\alpha_t} \odot s \odot i_t \odot x_t$ | $s_t = \alpha_t \odot s_{t-1} + \sqrt{(1-\alpha_t)} \odot i_t \odot x_t$ |
| Longhorn | $\|S - S_{t-1}\|_F^2 + \|Sk_t - x_t\|_{\mathrm{diag}(\beta_t)}^2$ | $S_t = (1_{m\times n} - \varepsilon_t \otimes k_t^{\odot 2}) \odot S_{t-1} +$ $(\varepsilon_t \odot x_t) \otimes k_t, \ \ \varepsilon_t = \beta_t/(1 + \beta_t k_t^\top k_t)$ |

Table 5: Some of the existing SSMs and their corresponding online learning objectives/updates.

## C  Proof

This section provides the proof for Theorem 3.1. Given the Longhorn's objective $S_t = \arg\min_{S \in \mathbb{R}^{d \times m}} \left\{ \|S - S_{t-1}\|_F^2 + \|Sk_t - x_t\|_{\mathrm{diag}(\beta_t)}^2 \right\}$, we have the following theorem:

**Theorem C.1.** *The closed form solution for $S_t$ for objective in Equation 5 is*

$$S_{t,i} = (I - \varepsilon_{t,i}k_t k_t^\top)S_{t-1,i} + \varepsilon_{t,i}k_t x_{t,i}, \quad \text{where } \varepsilon_{t,i} = \frac{\beta_{t,i}}{1 + \beta_{t,i}k_t^\top k_t} \in [0, \infty). \tag{9}$$

*Proof.* As the objective in equation 5 is in a quadratic form with respect to $s$, there is a unique minimum. Observe that each row of $S$ (e.g., $S_i$) optimizes the objective independently, therefore we can solve the solution row-wise. By setting the derivative of $\nabla_{S_i}L_t = 0$, we have:

$$\begin{aligned}
\nabla_{S_i}L_t = 0 &\iff (S_i - S_{t-1,i}) + \beta_{t,i}(S_i^\top k_t - x_{t,i})k_t = 0 \\
&\iff (I + \beta_{t,i}k_t k_t^\top)S_i = S_{t-1,i} + \beta_{t,i}k_t x_{t,i} \\
&\underset{(3)}{\iff} S_i = \left(I - \frac{\beta_{t,i}}{1 + \beta_{t,i}k_t^\top k_t}k_t k_t^\top\right)S_{t-1,i} + \left(I - \frac{\beta_{t,i}}{1 + \beta_{t,i}k_t^\top k_t}k_t k_t^\top\right)\beta_{t,i}k_t x_{t,i} \\
&\iff \left(I - \frac{\beta_{t,i}}{1 + \beta_{t,i}k_t^\top k_t}k_t k_t^\top\right)S_{t-1,i} + \frac{(I + \beta_{t,i}k_t^\top k_t - \beta_{t,i}k_t k_t^\top)\beta_{t,i}k_t x_{t,i}}{I + \beta_{t,i}k_t^\top k_t} \\
&\underset{(5)}{\iff} \left(I - \frac{\beta_{t,i}}{1 + \beta_{t,i}k_t^\top k_t}k_t k_t^\top\right)S_{t-1,i} + \frac{\beta_{t,i}k_t x_{t,i}}{I + \beta_{t,i}k_t^\top k_t}
\end{aligned}$$

(3) is derived from the fact that $(I + \beta_{t,i}k_t k_t^\top)^{-1} = (I - \frac{\beta_{t,i}k_t k_t^\top}{1 + \beta_{t,i}k_t^\top k_t})$ by the Sherman–Morrison formula. (5) is derived by noticing that $k_t^\top k_t k_t x_{t,i} - k_t k_t^\top k_t x_{t,i} = 0$.  □

## D  Additional Experiment Details

We provide the architecture detail for conducting the scaling law experiments on OpenWebText in Table 6. The architecture configs follow exactly from the Mamba paper (Gu & Dao, 2023).

| Params | n_layers | d_model | n_heads / d_head | Training steps | Learning Rate | Batch Size | Tokens |
|---|---|---|---|---|---|---|---|
| 125M | 12 | 768 | 12 / 64 | 4800 | 6e-4 | 0.5M tokens | 2.5B |
| 350M | 24 | 1024 | 16 / 64 | 13500 | 3e-4 | 0.5M tokens | 7B |

Table 6: Training details on OpenWebText.

