# OpenReview forum: "Longhorn: State Space Models are Amortized Online Learners"
_ICLR.cc/2025/Conference — ICLR 2025 Poster_

### Official Review · Reviewer_DVh4 · 2024-11-02

**Soundness:** 2
**Presentation:** 3
**Contribution:** 2
**Rating:** 6
**Confidence:** 4

**Summary:**

This paper presents a novel perspective on SSM models through the lens of online learning, offering a fresh analytical framework. The learning process consists of two losses: one ensuring minimal state updates, and another optimizing the reconstruction of current x from the state using key k. Within this framework, the authors propose Longhorn, which introduces differential importance weighting for various dimensions during reconstruction. Experimental results across different scales, tasks, and sequence lengths demonstrate Longhorn's superior compression rate on PassKey tasks and better length generalization, while maintaining comparable performance with baseline SSMs on other tasks. While the theoretical framework shows significant value, the paper makes an approximation step without thoroughly discussing its impact on the overall theory, which intuitively could lead to substantial differences.

**Strengths:**

- Novel theoretical perspective analyzing SSMs through online learning
- Proposed theoretical framework shows strong potential for length generalization
- Comprehensive empirical validation across various settings

**Weaknesses:**

- Insufficient discussion of the approximation's impact, creating a gap between theory and practice
- Limited comparison with contemporary methods (e.g., Mamba2 and DeltaNet); given the concurrent timing with DeltaNet, would appreciate author response on this comparison

**Questions:**

- There appears to be a discrepancy between the significant improvements in PPL versus the modest gains in downstream task metrics. Could the authors elaborate on this phenomenon?
- Could the authors provide analysis on why DeltaNet struggles with extrapolation, while Longhorn demonstrates superior extrapolation capabilities, especially given their similarities in the update equation?
- Is there an ablation study on the beta parameters in the OCP formula? What guidelines exist for its optimal selection?

---

> ### Author Response · Authors · 2024-11-25
> **Author Response**
>
> We sincerely thank the reviewer for the valuable suggestions and thoughtful questions. Below, we address each of your concerns in detail.
>
> **1. Insufficient discussion of the approximation's impact, creating a gap between theory and practice.**
>
> We acknowledge that the diagonal approximation introduces some discrepancies. However, to the best of our knowledge, there is currently no efficient method to implement the exact closed-form solution. Notably, the diagonal approximation aligns with the implementation of the Mamba model, ensuring direct comparability and avoiding additional overhead. In principle, an efficient matrix pscan algorithm could enable the exact closed-form implementation, which we leave as a direction for future work.
>
>
> **2. Limited comparison with contemporary methods (e.g., Mamba2 and DeltaNet); given the concurrent timing with DeltaNet, would appreciate author's response on this comparison.**
>
> We acknowledge that Mamba2 and DeltaNet are concurrent. Due to the limited resources we have and the short period of time in rebuttal, we cannot train DeltaNet. But please consider the following two remarks:
>
> - Mamba2’s main change to Mamba is to reduce the richness of the recurrence, and in return make the state size larger. Given this, if the state sizes are the same, Mamba should be strictly better than Mamba2. In addition, take Figure 9 in Mamba2 as an example, one can see that even though Mamba’s state size is much smaller than Mamba2, its scaling performance is quite the same as Mamba2 (indicating the importance of the richness of the recurrence form). Moreover, recent works like Jamba-1.5 [1] observes that in hybrid architecture that interleaves self-attention and SSMs, Mamba’s performance is much better than Mamba2. So we can expect that the A (state transition matrix) needs to be rich enough to bring extra benefit to self-attention.
>
> - Regarding DeltaNet, we find that it shares a similar state size as Mamba2, hence is much larger than Mamba’s state size. In the meantime, as the 100B subset of Slimpajama dataset DeltaNet uses and we use are different, it is hard to compare against DeltaNet fairly. Note that our Mamba model’s performance is better than the results in DeltaNet’s paper, which was taken from the GLA paper. From the method perspective, we want to argue that the contributions of DeltaNet and ours are different. DeltaNet focuses on computation efficient training (parallelization) of the original DeltaNet work. Both the original one and the more efficient DeltaNet require certain normalization techniques to make sure the A matrix is stable. In addition, as seen in the last paragraph of Section 5.3 in the more recent DeltaNet paper [2], the model does not extrapolate well beyond the training context. In comparison, Longhorn successfully extrapolates beyond its training context, which closely aligns with the original motivation that it does online learning. And Longhorn’s update ensures that the A matrix is always stable (absolute value of the eigenvalues is smaller than 1). Moreover, Longhorn is not only an architecture, but is also a framework. Future works can explore different online learning objectives, instead of directly working on the recurrence form.
>
> **3. There appears to be a discrepancy between the significant improvements in PPL versus the modest gains in downstream task metrics. Could the authors elaborate on this phenomenon?**
>
> Thanks for raising this point. The downstream evaluation metrics essentially are not what the model optimizes for, hence lower loss doesn’t always mean better metrics. But Longhorn indeed performs better in terms of both. On the other hand, as the 1B model is still small, the downstream evaluation metrics are not comparable to larger models, we expect those metrics will be more meaningful when actual large models are trained. In that regime, often the validation loss aligns better with the downstream metrics, and we expect that Longhorn’s advantage will be more significant.

---

> > ### Author Response · Authors · 2024-11-25
> > **Author Response (2)**
> >
> > **4. Could the authors provide an analysis of why DeltaNet struggles with extrapolation, while Longhorn demonstrates superior extrapolation capabilities, especially given their similarities in the update equation?**
> > There are two possibilities:
> > - One possibility is that: DeltaNet puts a normalization term on the key vector to prevent the A matrix from being unstable, this might cause some issues. In comparison, Longhorn’s A is derived directly from the closed form, and there is no need to conduct extra normalization.
> > - The other possibility is that: Longhorn, like Mamba, leverages the beta terms as a vector, which essentially performs d SSMs in parallel, but DeltaNet uses per-head beta terms. But it is also because of that that DeltaNet can leverage matrix multiplication for parallelization.
> > We are not sure which one is the cause or if it is a mix of both.
> >
> > **5. Is there an ablation study on the beta parameters in the OCP formula? What guidelines exist for its optimal selection?**
> >
> > Thanks for asking. Note that beta parameters are learned. There are really no hyperparameters regarding the SSM (in fact no more hyperparameters than Transformer), which is one of the main advantages of Longhorn, compared to many existing SSMs like Mamba. From this perspective, Longhorn is not only theoretically elegant but also simple to implement in practice.
> >
> >
> >
> > **References:**
> >
> > [1] Jamba-1.5: Hybrid Transformer-Mamba Models at Scale. https://arxiv.org/pdf/2408.12570
> >
> > [2] Linear Transformers Are Secretly Fast Weight Programmers. https://arxiv.org/abs/2102.11174

---

> ### Comment · Reviewer_DVh4 · 2024-11-28
>
> Thanks for your detailed response. Regarding the approximation issue, I admit that approximations can lead to more efficient implementations, but intuitively, this approximation may cause significant changes to the theoretical framework. One compromise approach is to compare the models before and after approximation on a small scale. Otherwise, it would be difficult to properly validate the value of both the theoretical framework and the approximation operation.

---

> > ### Author Response · Authors · 2024-12-04
> > **Response to Reviewer's Followup Comments**
> >
> > We appreciate the reviewer's insightful comments on the approximation issue. Currently, efficient and accurate implementation of matrix parallel scan kernels, which are essential for validating Longhorn’s exact update form, to our knowledge, is not feasible. This limitation extends even to small-scale language modeling tasks, where CUDA is necessary.
> > To this end, we write a simple associative recall Python script, use all recurrent forms of SSMs to compute the update, and use backpropagation through time (BPTT) to train the network (this is extremely inefficient, but is okay for small models on small problems). Here, we just use a single layer of self-attention, longhorn SSM, longhorn SSM (exact form), and GLA. The training loss and recall rate over training steps are provided in this link (https://anonymous.4open.science/r/longhorn_rebuttal-54F5/longhorn_exact_form_comparison.png)
> >
> > According to the plot, we can see that the Longhorn SSM (exact-form) is indeed better than Longhorn, and even much better than self-attention in this toy associative recall problem. But note that Longhorn SSM still outperforms Gated Linear Attention (GLA), which is consistent across all experiments in our paper. This result, though on a very small toy problem, indicates that there is a potential for the exact form of Longhorn once it is possible to implement the matrix parallel scan (pscan) efficiently and accurately. The authors have, in fact, attempted to implement the matrix pscan before but the written cuda kernel is numerically unstable and cannot be used for training, therefore we ended up with this diagonal approximation and it turned out to work well as well.
> >
> > Meanwhile, we would like to emphasize that the Longhorn paper not only presents this particular form of SSM but also provides a new perspective to design more powerful state space models as alternatives to self-attention models. We hope the insight from this work can inspire future research in this area.

---

### Official Review · Reviewer_FTDh · 2024-11-04

**Soundness:** 3
**Presentation:** 3
**Contribution:** 3
**Rating:** 6
**Confidence:** 3

**Summary:**

The paper introduces Longhorn, a novel state-space model (SSM) architecture designed as a meta-module that effectively handles sequence modeling problems. It described a theoretical framework based on online learning principles to derive the closed-form solutions for the online associative recall problem.

The empirical results convincingly demonstrate that Longhorn surpasses other state-of-the-art SSMs in performance, particularly highlighted by its impressive recall capabilities on the Multi-Query Associative Recall (MQAR) benchmark.

**Strengths:**

- the online learning framework provides a fair theoretical underpinning for understanding the Linear attention model / SSMs. This approach not only supports the conceptual innovations presented but also enhances the interpretability of SSM behaviors in practical applications.

- emirpical results: Longhorn has good sample efficiency compared to STOA models such as Mamba and GLA. This advantage is critical in scenarios where computational resources are limited.

**Weaknesses:**

Appoximation: while the diagonal approximation is a key aspect of Longhorn's implementation, its impact on the theoretical framework's alignment with empirical results remains unclear to me. I would expect a deeper exploration into how this approximation influences model performance could bridge the gap between theoretical predictions and observed outcomes.

**Questions:**

1. can you provide more details on the sample efficiency experiments? Say, what kinds of hyper-parameters did you try? Can you do an abaltion study?

2. Echoing the weakness of the paper, it is unclear to me that after using such an appoximation, is the theory framework still well aligned to the experiments?

---

> ### Author Response · Authors · 2024-11-25
> **Author Response**
>
> We sincerely thank the reviewer for the valuable suggestions and thoughtful questions. Below, we address each of your concerns in detail.
>
> **1. Approximation: while the diagonal approximation is a key aspect of Longhorn's implementation, its impact on the theoretical framework's alignment with empirical results remains unclear to me. I would expect a deeper exploration into how this approximation influences model performance could bridge the gap between theoretical predictions and observed outcomes.**
>
> We appreciate the reviewer’s observation. However, to the best of our knowledge, the exact closed-form SSM update cannot be efficiently implemented, preventing us from directly evaluating its modeling performance. The current approximation is the most efficient implementation we could identify that closely aligns with the original formulation. It also retains the same parallel structure as the Mamba architecture, ensuring a fair comparison. Despite this approximation, we observed that Longhorn improves sample efficiency in terms of perplexity compared to Mamba, suggesting that its inductive bias benefits language modeling. Implementing the exact closed form would require matrix pscan, which we have explored; however, it is currently numerically unstable and significantly slower.
>
> **2. Can you provide more details on the sample efficiency experiments? Say, what kinds of hyper-parameters did you try? Can you do an ablation study?**
>
> Given the size of the model and training, we did **not** have the resources to do a hyperparameter sweep or sensitivity analysis.  We use the same hyperparameters as Mamba.
>
>
> **3. Echoing the weakness of the paper, it is unclear to me that after using such an approximation, is the theory framework still well aligned to the experiments?**
>
> It is true that there is this discrepancy due to the approximation, however, this is the closest approximation we can find that has an efficient parallel form. As diagonal approximations of matrices have been widely used in optimization and other fields of deep learning for efficiency, we think this diagonal approximation is reasonable. The experiments also suggest improved sample efficiency even using the diagonal approximation, hence we find that it  indicates that Longhorn benefits from the inductive bias.  One way of thinking of it is that Longhorn is a practical method that is inspired by a theoretical formulation.  This discrepancy between theory and practice is not uncommon, and indeed is often inevitable as in this case.

---

### Official Review · Reviewer_LqBo · 2024-11-04

**Soundness:** 3
**Presentation:** 3
**Contribution:** 3
**Rating:** 6
**Confidence:** 4

**Summary:**

This paper introduces Longhorn, a new state-space model (SSM) architecture. By adopting an online learning optimization perspective, the authors unify several popular SSM architectures, bringing clarity to the structural differences between them. The explanation in Appendix A and Table 4 are especially helpful for understanding these nuances. Building on this unified approach, the authors propose a simplified SSM structure through a novel value retrieval mechanism based on key structures, offering insightful explanations of their method. The paper concludes by deriving a closed-form update formula for the state S in the SSM, supported by effective empirical results.

**Strengths:**

1. The paper is well-written and easy to follow. The clarity of explanation makes complex ideas accessible, particularly in sections like Appendices A and B, which provide valuable insights into the nuances of different approaches.
2. The novelty of the new formulation for the Longhorn approach is impressive. The retrieval-based perspective is both innovative and elegantly presented, offering a fresh solution that enhances the field.
3. The exploration of SSM structure variances through online learning optimization is also intriguing and adds depth to the paper’s contribution.

**Weaknesses:**

1. While this paper presents a focused study on architecture, the data and model scale seem limited. Expanding the experimental scale and providing a more comprehensive analysis would significantly enhance the paper's impact.
2. The reduction in perplexity compared to Mamba is notable. However, the results in Table 2 appear mixed, which could benefit from further clarification or exploration.
3. Including additional experiments, such as MMLU, GSM-8K, and more extensive long-context benchmarks, would strengthen the findings and provide a more robust evaluation of the model's capabilities.

**Questions:**

see weakness.

---

> ### Author Response · Authors · 2024-11-25
> **Author Response**
>
> We sincerely thank the reviewer for the valuable suggestions and thoughtful questions. Below, we address each of your concerns in detail.
>
> **1. While this paper presents a focused study on architecture, the data and model scale seem limited. Expanding the experimental scale and providing a more comprehensive analysis would significantly enhance the paper's impact.**
>
> We agree with the reviewer on this. However, due to the limited computation resources we have in our lab, the current experiments have already taken us several months to run. We are happy to experiment on larger models/dataset once we have more compute in the future.
>
> **2. The reduction in perplexity compared to Mamba is notable. However, the results in Table 2 appear mixed, which could benefit from further clarification or exploration.**
>
> Yes, the reduction in perplexity indicates that Longhorn benefits from its inductive bias (in terms of language modeling). We think the downstream evaluations might not directly reflect this as they are noisy evaluation metrics of a given model, and since we are only comparing 1B size models, they might not differ too much from each other.
>
>
> **3. Including additional experiments, such as MMLU, GSM-8K, and more extensive long-context benchmarks, would strengthen the findings and provide a more robust evaluation of the model's capabilities.**
>
> We appreciate the reviewer’s suggestion to include evaluations on benchmarks like MMLU and GSM-8K. After investigating, we found that all 1B models in our experiments yield near-random performance on these benchmarks. For reference, even LLaMA 7B, trained on 1–2T tokens, achieves only ~25% accuracy on MMLU in the 0-shot setting (and 35% in 5-shot) (see Table 9 of [1]), which aligns with random guessing. Given that our models are 1B models trained on 100B tokens, their similar near-random performance is expected. Similarly, GSM-8K results are limited, as the SlimPajama dataset lacks sufficient high-quality mathematical reasoning data.
> However, we emphasize the compact design of Mamba and Longhorn, which have a state size of only 16—significantly smaller than that of GLA or self-attention. Despite this, Longhorn achieves superior performance on 4K context length tasks and outperforms Mamba on both synthetic and large-scale language modeling benchmarks. We believe Longhorn's direct approach to solving the online associative recall problem will demonstrate even greater advantages as context lengths increase (just as what is shown in the MQAR example).
>
> [1] LLaMA: Open and Efficient Foundation Language Models. https://arxiv.org/pdf/2302.13971

---

### Official Review · Reviewer_dxPc · 2024-11-04

**Soundness:** 3
**Presentation:** 4
**Contribution:** 3
**Rating:** 6
**Confidence:** 5

**Summary:**

Longhorn formulates state-space models by solving an online regression problem. By designing various online learning objectives, it can induce different linear recurrent models, providing a unified framework and principled approach for developing new models. This work adopts an objective that encourages the hidden state to remain close to its previous state (i.e., \( ||S_{t+1} - S_t||_F \)) and includes a term that promotes key-input association (i.e., \( ||S k_t - x_t||_{\text{diag}(\beta_t)^2} \)). While the optimal solution is derived, it presents computational challenges. To address this, a diagonal approximation is used, resulting in a model computationally similar to Mamba (and thus similarly efficient) but with improved empirical performance, as demonstrated on synthetic datasets and medium-scale language modeling and image modeling.

**Strengths:**

1. The paper is well-written and easy to follow.
2. The formulation via online learning for solving in-context associative recall is interesting and elegant. It explains why Longhorn (and also DeltaNet) performs well in MQAR tasks.
3. Empirical results look good.

**Weaknesses:**

1. The main issue with this work is that the implementation does not fully align with the theory. Using a diagonal matrix to approximate an “identity-plus-low-rank” dense matrix is coarse, and it’s unclear if the theoretical advantage translates to this setting.
2. In Eq5, the norm \(\text{diag}(\beta_t)\) appears unusual and is not well-motivated or empirically validated. Why is a vector-valued \(\epsilon\) necessary? If not, the DeltaNet structure could leverage the kernel from Yang et al. (2024, https://arxiv.org/abs/2406.06484), which would easily scale up the head dimension and likely benefit recall-intensive tasks requiring a large state size. Longhorn, as it stands, cannot be expressed in matmul form, leading to similar challenges as in Mamba. Would Mamba2-like optimization, potentially resulting in a DeltaNet-like model with scalar \(\epsilon\), be preferable?
3. MQAR is a synthetic dataset and insufficient to demonstrate Longhorn’s advantages in recall-intensive tasks. Results on real-world recall-intensive tasks proposed in Arora 2024 [https://arxiv.org/abs/2402.18668] would provide a stronger case. Could you report zero-shot accuracy on these tasks? A table similar to [Table 1, https://arxiv.org/abs/2407.05483] would be very useful and necessary.
4.  This work lacks several ablation studies. For instance, the "value" projection is removed compared to standard models, yet this change is not analyzed. Additionally, the model does not clarify the benefits of parameter tying.

**Questions:**

Are there any actual wall-time comparisons in terms of training & inference?

---

> ### Author Response · Authors · 2024-11-25
> **Author Response**
>
> We sincerely thank the reviewer for the valuable suggestions and thoughtful questions. Below, we address each of your concerns in detail.
>
> **1. The main issue with this work is that the implementation does not fully align with the theory. Using a diagonal matrix to approximate an “identity-plus-low-rank” dense matrix is coarse, and it’s unclear if the theoretical advantage translates to this setting.**
>
> We acknowledge that the diagonal matrix is an approximation. However, to the best of our knowledge, there is currently no efficient method to implement the exact closed-form solution. Notably, the diagonal approximation aligns with the implementation of the Mamba model, ensuring direct comparability and avoiding additional overhead. In principle, an efficient matrix pscan algorithm could enable the exact closed-form implementation, which we leave as a direction for future work.
>
>
> **2. In Eq5, the norm (\text{diag}(\beta_t)) appears unusual and is not well-motivated or empirically validated. Why is a vector-valued (\epsilon) necessary? If not, the DeltaNet structure could leverage the kernel from Yang et al. (2024, https://arxiv.org/abs/2406.06484), which would easily scale up the head dimension and likely benefit recall-intensive tasks requiring a large state size. Longhorn, as it stands, cannot be expressed in matmul form, leading to similar challenges as in Mamba. Would Mamba2-like optimization, potentially resulting in a DeltaNet-like model with scalar (\epsilon), be preferable?**
>
> Having $\beta$ as a vector is reasonable because, like Mamba, we treat each dimension of $x$ as a separate state space model. This approach enhances the sequence model's representational power. Since each dimension of $x$ can have its own $\beta$, it is natural to represent $\beta$ as a vector, leading to the $(\text{diag}(\beta_t))$ norm. However, we agree with the reviewer that this introduces the limitation that, similar to Mamba, Longhorn cannot be expressed in matmul form. If $\beta$ were instead shared across all dimensions of $x$, Longhorn could be implemented in the same way as DeltaNet.
>
>
> **3. MQAR is a synthetic dataset and insufficient to demonstrate Longhorn’s advantages in recall-intensive tasks. Results on real-world recall-intensive tasks proposed in Arora 2024 [https://arxiv.org/abs/2402.18668] would provide a stronger case. Could you report zero-shot accuracy on these tasks? A table similar to [Table 1, https://arxiv.org/abs/2407.05483] would be very useful and necessary.**
>
> We thank the reviewer for the suggestion. However, as Table 1 in the paper mentioned, the authors trained 1B Mamba over 300B tokens but we train Longhorn only using 100B tokens, it would not make a fair comparison. Given the short period of time during rebuttal, we could not retrain another Longhorn model over 300B tokens (it might take several weeks). Additionally, we would like to point out that to compare different state space models’ recall ability really fairly, one would make the state size the same. From this perspective, Mamba and Longhorn both have the smallest state sizes across all SSMs in most recall comparison experiments (even including MQAR). We hypothesize that they would have even better recall rates if Mamba and Longhorn were scaled to larger state sizes.  We will add this evaluation as an interesting direction for future work.
>
>
> **4. This work lacks several ablation studies. For instance, the "value" projection is removed compared to standard models, yet this change is not analyzed. Additionally, the model does not clarify the benefits of parameter tying.**
>
> We did not omit the value projection, as we aimed for a fair comparison with Mamba, which uses the same parameterization except for the updating functions. To ensure consistency, we followed the Mamba architecture exactly: in Mamba, $v$ is first projected, and $q$ and $k$ are linearly projected from $v$. We adopted the same approach, modifying only the SSM component, as shown in Figure 3. Additionally, we clarified in Line 9 of Algorithm 1 (Lines 123–125) that $x_t$ is preprocessed through a linear projection followed by a 1D convolution, where the resulting $x_t$ serves as the value.
>
> **5. Wall-clock time comparison.**
>
> Unfortunately, we did not keep track of the wall-clock training time. But we did some offline kernel speed tests and Mamba and Longhorn achieved almost the same speed (0.122s for Mamba and 0.124s for Longhorn, which differs within 2% in performance). And the Mamba and Longhorn architecture is exactly the same except for the SSM kernel, hence the overall training/inference speeds are also nearly the same.

---

> ### Comment · Reviewer_dxPc · 2024-11-26
>
> Since most of my concerns are not addressed, I temporarily decrease my score to 5.
>
> > Having $\beta$ as a vector is reasonable because, like Mamba, we treat each dimension of as a separate state space model. This approach enhances the sequence model's representational power.
>
> Is vector-valued $\beta$ really useful in practice? Could you please add some ablation studies to support your claim? Training a small scale Longhorn model with $\beta$ sharing across all dimensions in $x$ using the DeltaNet kernel should suffice.
>
> > However, as Table 1 in the paper mentioned, the authors trained 1B Mamba over 300B tokens but we train Longhorn only using 100B tokens, it would not make a fair comparison. Given the short period of time during rebuttal, we could not retrain another Longhorn model over 300B tokens (it might take several weeks)
>
> I am not suggesting you to retrain the Longhorn model, but just to add evaluations in real-world recall-intensive tasks of your models trained in the Table 2 of your paper. Tasks such as FDA, SWDE, NQ, SQUAD, etc. mentioned in [1] should be good.
>
> > We did not omit the value projection, as we aimed for a fair comparison with Mamba, which uses the same parameterization except for the updating functions.
>
> Thanks for your clarification, but you mentioned in L93-94, "Thus Longhorn does not need a separately parameterized forget gate, which saves parameters when the state size is large." In this case, you need separately parameterized beta, right? Are total parameters the same then? Can you comment on this claim in L93-94?
>
> ---
> [1] Just read twice: closing the recall gap for recurrent language models (arXiv 2024)

---

> ### Author Response · Authors · 2024-12-04
> **Response to Reviewer's Followup Question**
>
> We thank the reviewer for their follow-up on the concerns and feel sorry for the late response as we were preparing the experiment results. Here are our latest results and response and hope those could address your concern.
>
>
> **1. Is vector-valued $\beta$ useful in practice?**
>
> To verify that the vector value $\beta$ is useful, we train a 120M size Longhorn but keep the $\beta$ term a learnable scalar. So everything is the same, including the update form. The only change is that we broadcast the scalar $\beta$ to all dimensions. The experiment setting follows exactly as in Section 5.2 of the paper, where we use a 1024 context length.
>
> The results are the following:
>
>
> | Model                        | Validation Loss |
> |------------------------------|-----------------|
> | Longhorn (with scalar $\beta$)     | 3.262           |
> | Longhorn (with vector $\beta$)     | 3.225           |
> | Mamba                        | 3.238           |
> | LLaMA                        | 3.247           |
> | GLA                          | 3.381           |
>
>
> From this, we conclude that the performance of  Longhorn with scalar $\beta$ is much worse than that of Longhorn with vector $\beta$.
>
>
> **2. Real-world recall intensive benchmark.**
>
> We thank the reviewer for the clarification of the concern. We have evaluated the 1B models we trained on the suggested benchmark. Here are the results we got for Longhorn-1B and Mamba-1B.
>
> | Model     | FDA        | SWDE      | NQ        | SQuAD     | TriviaQA  | Drop      | AVG        |
> |-----------|------------|-----------|-----------|-----------|-----------|-----------|------------|
> | Mamba     | 33.2/40.6  | 35.0/36.2   | 26.6/32.6 | 37.4/52.7 | 56.3/56.9 | 20.4/31.5 | 34.82/41.75 |
> | Longhorn  | 40.2/50.4  | 33.2/42.3 | 27.6/33.2 | 35.0/55.0     | 58.5/55.9 | 21.3/33.3 | 35.97/45.0 |
>
> From the results, we observe that Longhorn achieved a significant 1.15 (4.25 if read twice) improvement over Mamba. Note that there is a gap between the Mamba 1B in the suggested paper and ours here. The reason might be due to: 1. we trained for 100B instead of 300B tokens; 2. we used SlimPajama instead of Pile.
>
> We will include this additional result in our paper in the final version. Thanks for suggesting this benchmark.
>
> **3. Longhorn saves parameters.**
>
> For Mamba’s SSM module, it requires two linear projections to the B and C matrices (in our case, the K and Q matrices). It also requires one (bottlenecked) linear projection to compute the $\Delta \in \mathbb{R}^d$ term (the step size), this is equivalent to our $\beta \in \mathbb{R}^d$. However, Mamba additional requires an input-independent A matrix $A \in \mathbb{R}^{m \times d}$, which Longhorn does not require. Therefore, we say that Longhorn has slightly fewer parameters than Mamba, and the gap is proportional to $\mathcal{O}(md)$.
>
> To further convince the reviewer that the inductive bias from Longhorn is useful, in one of our old experiments, we compared Longhorn against a Mamba’s variant that is more similar to Longhorn, where we made Mamba’s $A$ matrix, instead of a constant learnable matrix, also an input dependent vector. So $A = W_A x_t \in \mathbb{R}^m$ and $W_a \in \mathbb{R}^{m \times d}$. We call this Mamba (with learnable $A$). The final update rule of Mamba (with learnable A) is:
>
> $$S_t = \exp( -\Delta \otimes A_t ) \odot S_{t-1} + (\Delta \odot x_t) \otimes B_t$$
>
> In comparison, Longhorn’s update is
>
> $$S_t = (1_{d \times m} - \epsilon_t \otimes k_t^{\odot 2}) \odot S_{t-1} + (\epsilon_t \odot x_t) \otimes k_t $$
>
> Here, $S_t \in \mathbb{R}^{d \times m}$. $k_t$ matches the size of $B_t$ in Mamba, $\epsilon_t$ matches the size of $\Delta_t$ in Mamba. Note that the main difference is that in Longhorn, the forget gate $k_t^{\odot 2}$ is linked to the input gate $k_t$, while in Mamba (with learnable $A$) the two are separate.
>
> The old experiment setting used a 350M model and 1024 context length (also the same experiment setting as in Section 5.2), the results are the following:
>
>
> | Model                        | Validation Loss |
> |------------------------------|-----------------|
> | Longhorn (with vector $\beta$)     | 2.888           |
> | Mamba                        | 2.902           |
> | LLaMA                        | 2.891           |
> | GLA                          | 3.018           |
> | Mamba (with learnable $A$)     | 2.922           |
>
>
> From this, we see that with this learnable forgetting gate $A = W_A x_t$, the performance of Mamba (with learnable $A$) is even worse than the original Mamba. We do not know the exact reason behind this, but given the similarity between the recurrent update rule of Mamba (with learnable $A$) and that of Longhorn, we believe it indicates the importance of the inductive bias in Longhorn. Namely, the forgetting is roughly about the square of the input might be beneficial (i.e., when $k$’s magnitude < 1, which is usually the case, it means Longhorn forgets less but inputs more).

---

### Meta-Review · Area_Chair_1uPT · 2024-12-22

**Metareview:**

This paper offers a novel online-learning perspective on SSM design, introducing a novel architecture called Longhorn and demonstrating improved performance over strong baselines like Mamba. All reviewers appreciated the clarity of exposition, the elegance of the theoretical framework, and the thorough empirical comparisons showing Longhorn’s effectiveness in tasks requiring long-context processing. While the reviewers initially expressed some concerns (e.g., diagonal approximation), they later found the rebuttal and additional experiments convincing.

**Additional Comments On Reviewer Discussion:**

While the reviewers initially expressed some concerns (e.g., diagonal approximation), they later found the rebuttal and additional experiments convincing.

---

### Decision · Program_Chairs · 2025-01-22

Accept (Poster)